# A navigational risk evaluation of ferry transport: Continuous risk management matrix based on fuzzy Best-Worst Method

**Linh Thi Pham[1], Long Van Hoang**[2]*****

1 Faculty of Accounting—Finance, Dong Nai Technology University, Bien Hoa City, Vietnam, 2 Faculty of Management, Ho Chi Minh University of Law, Ho Chi Minh City, Vietnam

* longvanhoang1976@gmail.com, hvlong@hcmulaw.edu.vn

**Data Availability Statement:** All relevant data are within the manuscript and its Supporting Information files.

## Abstract

Ferry transport has witnessed numerous fatal accidents due to unsafe navigation; thus, it is of paramount importance to mitigate risks and enhance safety measures in ferry navigation. This paper aims to evaluate the navigational risk of ferry transport by a continuous risk management matrix (CRMM) based on the fuzzy Best-Worst Method (BMW). Its originalities include developing CRMM to figure out the risk level of risk factors (RFs) for ferry transport and adopting fuzzy BWM to estimate the probability and severity weights vector of RFs. Empirical results show that twenty RFs for ferry navigation are divided into four zones corresponding to their risk values, including extreme-risk, high-risk, medium-risk, and low-risk areas. Particularly, results identify three extreme-risk RFs: inadequate evacuation and emergency response features, marine traffic congestion, and insufficient training on navigational regulations. The proposed research model can provide a methodological reference to the pertinent studies regarding risk management and multiple-criteria decision analysis (MCDA).

## 1. Introduction

It has been argued that ferry transport is playing a more and more critical role in the economic development of countries, especially nations having long coastlines. More particularly, ferries contribute considerably to regional integration and accessibility and, in turn, provide a cost-effective means of transporting goods and services [1, 2]. Additionally, cruise ferries often serve as a scenic and enjoyable mode of transportation for tourists, thus facilitating the development of tourism in coastal areas [3, 4]. However, the safety of ferry transportation has attracted much concern from governments. On top of that, recent accidents necessitate a comprehensive approach to mitigate ferry navigation-related risks.

Recently, ferry transport has witnessed numerous fatal accidents due to unsafe navigation. According to Golden and Weisbrod [5], about 232 ferry incidents occurred between 2000 and 2014 in 43 countries, with a total of 21,574 fatalities appearing, averaging 130 deaths per incident and 1,541 deaths annually. In general, developing countries experienced 94% of total

**Funding:** The author(s) received no specific funding for this work.

**Competing interests:** The authors have declared that no competing interests exist.

accidents and 97% of total fatalities. Besides, some primary reasons for such disasters were reported, such as gas explosions [6], fire [5], capsizing [7], propulsion system malfunctions [8], misjudging distances [1], etc. Vietnam Inland Waterways Administration (VIWA) officially declared that 679 navigational accidents in terms of ferry transport happened between 2014 and 2020 by some leading causes, for example, mistakes in navigation (13.21%), crew distraction (51.07%), inadequate maintenance (23.21%), overcrowding (4.99%), and propulsion system malfunctions (11.44%). Thus, these occurrences not only emphasize the susceptibility of ferry transportation but also stress the crucial necessity for mitigating risks and consistently enhancing safety measures in ferry navigation.

According to Aven [9], a traditional risk management matrix (TRMM) has been adopted extensively to assess risks in ferry transport, because it allows for the quantification of risks based on their potential impact and probability of occurrence. By using TRMM, risk managers can assign qualitative or quantitative values to the likelihood (or probability) and severity (or consequence) of specific risks. These values are then plotted on a matrix to visualize the level of risk associated with different scenarios. Nonetheless, values assigning specific risks are discrete numbers (i.e., 1~5); thus, the weight of probability and severity are discontinuous values. For that reason, TRMM has some drawbacks, such as weak consistency [10], betweenness [11], and consistent coloring [12]. Therefore, the concept of the continuous risk management matrix (CRMM) is proposed to overcome this shortcoming.

Moreover, risk evaluation in ferry transport is characterized as a multi-criteria decision analysis (MCDA) problem [13]. In the case of ferry transport, where various factors contribute to the complexity of risk, MCDA allows decision-makers (DMs) to consider multiple criteria simultaneously when assessing risks. Nevertheless, some of the most common tools of MCDA, such as AHP, ANP, and SAW, require numerous pairwise comparisons (PCs) of risks, thereby not only weakening their practical application, but also increasing the inconsistency of PCs [14]. To cope with this challenge, the Best-Worst Method (BWM) developed by Rezaei [15] has been adopted extensively to solve MCDA. Compared with the classic tools of MCDA, the primary strength of BWM is fewer PCs, thus prone to obtaining DMs' judgment and boosting the consistency of subjective evaluation. Additionally, it is illustrated that DMs' subjective assessment is often uncertain and imprecise. Accordingly, the theory of the fuzzy set is incorporated into BWM to allow for the representation of degrees of such uncertainty and vagueness.

To sum up, motivations for this research is as follow. First, the safety of ferry transportation has attracted much concern from governments. On top of that, recent accidents necessitate a comprehensive approach to mitigate ferry navigation-related risks. Second, ferry transport has currently witnessed numerous fatal accidents due to unsafe navigation. Thus, these accidents not only emphasize the susceptibility of ferry transportation but also stress the crucial necessity for mitigating risks and consistently enhancing safety measures in ferry navigation. Third, TRMM has been adopted extensively to assess risks in ferry transport. Nonetheless, TRMM has some drawbacks that should be overcome. Therefore, CRMM is proposed to deal with this shortcoming. Fourth, compared with the classic tools of MCDA, the advantage of BWM is fewer PCs, thus increasing its application. Additionally, DMs' subjective assessment is arguably uncertain and imprecise that is coped with the fuzzy set theory.

To fill the literature gap, this current paper aims to carry out a navigational risk evaluation of ferry transport by the CRMM based on fuzzy BWM. To accomplish that, risk factors (RFs) affecting the risk of navigational safety for ferry transportation are first identified. Afterwards, fuzzy BWM is adopted to calculate the probability and severity of such RFs. Then, CRMM is constructed to rank RFs' risk value. Finally, some major ferry operators in Vietnam (the FO-VN case) are empirically surveyed to verify the proposed research model.

Key contributions of this study include the following:

- The current paper identifies five dimensions with twenty RFs for ferry navigation. By means of CRMM, RFs are divided into four zones corresponding to their risk values, including extreme-risk, high-risk, medium-risk, and low-risk areas.

- The application of fuzzy BWM in calculating the probability and severity of RFs can provide a methodological reference to the MCDA research. Compared with the pairwise comparison-based tools (i.e., AHP, ANP, and SAW), BWM does not require a full pairwise comparison matrix. In addition, integrating fuzzy theory into BWM allows a more realistic representation of uncertainty, vagueness, and imprecision.

- As an empirical study, the paper surveys three major FOs in Vietnam to verify the proposed research model. Results identify three extreme-risk RFs: inadequate evacuation and emergency response features, marine traffic congestion, and insufficient training on navigational regulations.

The subsequent parts of this paper are structured as follows: Section 2 presents the literature review of this study. Section 3 then elucidates the research methods. The case study is detailed in Section 4. Lastly, section 5 encompasses conclusions, limitations, and suggestions for future research directions.

## 2. Literature review

It has been argued that marine transport in general and ferry transport in particular are accident-prone sectors [16, 17]. Based on the extensive literature review and marine transport's features, the below section presents risk factors affecting the navigation safety of ferry transport.

Human factors are presumably indispensable in ferry transport; thus, various risks associated with them can significantly impact the safety and efficiency of maritime operations. Yuan, Wang [18] argued that crew fatigue regularly occurs in marine transport. They explained that ferry crews often work long hours and irregular schedules, thus affecting their cognitive function, reaction times, and decision-making abilities. According to Xue, Papadimitriou [8], fatigued crew members may struggle to respond effectively to unexpected situations, thereby increasing the likelihood of errors and accidents. Another risk factor impacting the safety of ferry transport is crew unfamiliarity with vessel systems. Aziz, Ahmed [19] demonstrated that inadequate knowledge of the ferry's intricate machinery and technology can result in operational errors in ferry navigation. Uğurlu, Yıldırım [20] pointed out that a lack of crew training and competence can compromise the crew's ability to navigate ships safely, and use onboard equipment effectively. Also, inadequate training increases the probability of human error, and potentially causing accidents [21]. Some prior studies agreed that communication breakdown is a pervasive risk factor in maritime transport, especially in emergency scenarios, where clear and precise communication is paramount. According to Nguyen, Ngo [1], misunderstandings in communication among crew members can have many severe consequences, such as navigation errors [22], maneuvering conflicts [23], and equipment operation mistakes [24]. One might conclude that by addressing these human factors and associated risks, FOs can enhance the overall safety and reliability of maritime transportation.

Similar to human factors, navigational equipment is also a critical component in ferry transport. Thence, its functioning-related risks can create many serious challenges to ferry safety. Hsu, Tai [7] postulated that the malfunction of electronic navigation systems (i.e., GPS and radar) occurs rather frequently in marine operations. Besides, using outdated navigational

charts in ferry transport may pose a substantial risk to maritime safety. It is evident nautical charts are indispensable tools for ensuring the safe passage of vessels through waterways, since they provide crucial information about the depth of water [8], the location of navigational hazards [25], and the configuration of the seabed [26]. Accordingly, the potential for navigational errors and incidents will significantly increase if these charts become inaccurate or outdated.

Additionally, limited visibility of navigational aids (i.e., buoys, lighthouses, and beacons) causes severe risks for ferry transport, especially during unfavorable weather conditions or low-light circumstances. Wang, Liu [27] illustrated that poor visibility can restrict crews from identifying important points at sea, thus compromising navigating safely through waterways. Moreover, many prior research agreed that the lack of redundancy and backup systems for navigational equipment in ferry transport represents a considerable vulnerability [28], and generates risks to maritime safety and operational continuity [29]. Wang, Liu [30] also explained that when a technical malfunction or failure in the primary navigational equipment happens, the absence of backup systems implies that FOs and crews do not have any other way to navigate vessels safely.

Navigation regulations are arguably the backbone of safe and efficient maritime transport. Thus, the disobedience of these regulations can cause numerous devastating risks to maritime navigation safety. Başhan, Demirel [31] pointed out that failure to adhere to internationally recognized rules, viz., COLREGs (International Regulations for Preventing Collisions at Sea), can lead to confusion and potentially dangerous situations, especially in high-traffic areas or during encounters with other vessels. Moreover, FOs must navigate through a maze of regulations; thus, a lack of understanding of local rules can result in navigational errors [32], increasing the risk of collisions and other incidents [33]. Further, Fan, Wang [34] demonstrated that without proper training on maritime regulations, the crew may find it challenging to interpret and respond to navigational signals. In addition, some previous research concluded that poor communication with maritime authorities further exacerbates the risks associated with navigation regulations [35].

The design of vessels in ferry transport is of paramount importance in ensuring the safety and effectiveness of maritime operations. First, inadequate stability and seakeeping characteristics, which are defined as a vessel's ability to remain at sea in all conditions and carry out its intended mission, can lead to the risk of capsizing, especially in the prevailing weather conditions and sea-state [36]. Another design-related risk is poor visibility from the bridge. It is highly admitted that navigating a ship under conditions of limited visibility is one of the most difficult challenges in ensuring a safe journey at sea, since it can increase the probability of a collision and grounding by two-fold [37, 38]. Unfortunately, this risk factor often happens in least-developed and developing nations, where the utilization of old and low-equipped ship in water transport is still common due to limited financial resources and the lack of regulatory frameworks and enforcement. Next, according to Akyuz [39], a ferry equipped with redundancy in propulsion systems may reduce the likelihood of system failures, thus ensuring its vital systems remain operational, even in the face of adversity. Additionally, the design of evacuation routes, life-saving equipment, and emergency response mechanisms must be carefully considered to ensure the swift and safe evacuation of passengers and crew during a crisis [40, 41].

Ferry transport is inherently influenced by external environmental factors that can introduce devastating risks to maritime operations. It has been argued that unpredictable weather patterns (i.e., storms, high winds, and rough seas) can cause many substantial challenges to ferry navigation, and increase risks of capsizing, collisions, and grounding [42, 43]. Besides, tidal and current conditions presumably present another set of risks, particularly in coastal and narrow waterways. Sys, Van de Voorde [44] explained that rapid changes in tidal flow and

strong currents can affect a ferry's maneuverability, making navigation more complex and increasing the probability of sea accidents. Further, according to Kulkarni, Goerlandt [45], floating debris and waterway obstacles can cause vessel damage and navigation hazards. More specifically, some debris (i.e., logs or containers) can damage the vessel's hull or propulsion systems [46], thereby causing operational disruptions and sea accidents [47]. Additionally, Xu, Ma [48] illustrated that marine traffic congestion in busy ports and narrow channels also introduces the risk of collisions and challenges in maintaining safe distances between vessels.

In conclusion, the navigational risk evaluation helps marine operators assess the hazards and risks affecting vessel navigation. Therefore, identifying RFs of marine transport is of paramount importance to guarantee navigation safety, thereby reducing potential accidents, and loss of lives and goods for the fast-ferry transportation. Table 1 also summarize the relevant literature.

## 3. Methods

### 3.1 Research framework

Fig 1 is the flowchart visually representing the process of this research study. After determining research objectives, the paper finds out risk factors of ferry transport navigation thanks to expert consultation and relevant literature. Then, fuzzy BWM is adopted to calculate severity and probability of risk factors. Next, CRMM is established to assess navigational risks of ferry transport. Ultimately, some policies are suggested to improve navigational risks of ferry transport.

### 3.2 Fuzzy Best-Worst Method

**3.2.1 Triangular fuzzy number.** Fuzzy set theory is a mathematical framework that can deal with problems of ambiguous, subjective and imprecise judgments [49, 50]. Extended from classical set theory, whose elements either belong to a set or do not, elements in fuzzy set theory can have degrees of membership $\mu_{\tilde{a}}(x)$ between 0 and 1, representing the degree to which an element belongs to a set.

**Definition 1**: $\tilde{a}$ is defined as a fuzzy number if its representation is given by: $\tilde{a} = \{(x, \mu_{\tilde{a}}(x)) \mid x \in \mathbb{R}\}$. Here, $(x, \mu_{\tilde{a}}(x))$ is an ordered pair, where $x$ is a real number and $\mu_{\tilde{a}}(x)$ is its degree of membership in the fuzzy number $\tilde{a}$.

The membership function $\mu_{\tilde{a}}(x)$ satisfies the following conditions:

1. $0 \leq \mu_{\tilde{a}}(x) \leq 1$ for all $x$ in the real number line.

2. $\mu_{\tilde{a}}(x)$ is a continuous function.

3. The support of $x$, denoted by $supp(x)$, is the set of all $x$ for which $\mu_{\tilde{a}}(x) > 0$.

4. The union of the supports of all elements in $x$ covers the entire real number line.

**Definition 2**: A fuzzy number $\tilde{a} \in \mathbb{R}$ is defined as a triangular fuzzy number (TFN) if its membership function is given by [51]:

$$\mu_{\tilde{a}}(x) = \begin{cases} 0, & x < l \\ \dfrac{x - l}{m - l}, & l \leq x < m \\ \dfrac{u - x}{u - m}, & m \leq x \leq u \\ 0, & x > u \end{cases}. \tag{1}$$

**Table 1. Summary of the relevant literature.**

| Challenges | References | Solutions | Strengths | Weaknesses |
|---|---|---|---|---|
| **1. Human Factors** | | | | |
| Crew fatigue | Yuan et al. (2021); Xue et al. (2021) | Improved scheduling and rest periods | Enhances cognitive function and decision-making | Requires significant operational changes |
| Crew unfamiliarity with vessel systems | Aziz et al. (2019); Uğurlu et al. (2015) | Comprehensive training programs | Reduces operational errors | Training can be time-consuming and costly |
| Communication breakdowns | Nguyen et al. (2022); Jon et al. (2021); Amro et al. (2020); Pan & Hildre (2018) | Standardized communication protocols | Improves response in emergencies | Implementation consistency can be challenging |
| **2. Navigational Equipment** | | | | |
| Malfunction of electronic navigation systems | Hsu et al. (2022) | Regular maintenance and updates | Ensures accurate navigation | Requires constant monitoring and resources |
| Outdated navigational charts | Xue et al. (2021); Mohammed et al. (2016); Hiremath et al. (2016) | Regular chart updates | Provides accurate navigation data | Dependent on timely updates |
| Limited visibility of navigational aids | Wang et al. (2019) | Enhanced visibility technology | Improves navigation in poor conditions | Technology may be expensive |
| Lack of redundancy in equipment | Wang & Chin (2016); Mia et al. (2021) | Implementing backup systems | Ensures operational continuity | Increases initial setup costs |
| **3. Navigation Regulations** | | | | |
| Disobedience of regulations | Başhan et al. (2020) | Strict enforcement and training | Ensures compliance and safety | Requires continuous monitoring |
| Lack of understanding of local rules | Ung (2018; Arof & Nair (2017) | Regular regulatory training | Reduces navigational errors | Training programs need regular updates |
| Poor communication with authorities | Baldauf & Hong (2016) | Improved communication systems | Enhances coordination | Implementation and maintenance can be difficult |
| **4. Vessel Design** | | | | |
| Inadequate stability and seakeeping | Ozturk & Cicek (2019) | Design improvements | Reduces capsizing risk | Design changes can be costly |
| Poor visibility from the bridge | Howe et al. (2016); Solomon et al. (2021) | Advanced design standards | Improves navigation safety | Upgrading old vessels can be difficult |
| Lack of redundancy in propulsion systems | Akyuz (2017) | Redundant systems in design | Ensures continuous operation | Increased design complexity |
| Ineffective evacuation routes | Ung (2021); Wood et al. (2018) | Improved emergency designs | Enhances safety during crises | Design changes can be expensive |
| **5. External Environmental Factors** | | | | |
| Unpredictable weather patterns | Cui (2019; X. Wang et al. (2021) | Advanced weather forecasting | Reduces risk of accidents | Forecasting technology may be limited |
| Tidal and current conditions | Sys et al. (2020) | Advanced navigation systems | Improves maneuverability | Technology can be costly |
| Floating debris and obstacles | Kulkarni et al. (2020); Chang et al. (2015); Abbassi et al. (2017) | Regular monitoring and clearing | Reduces navigation hazards | Requires continuous effort |
| Marine traffic congestion | Xu et al. (2020) | Improved traffic management | Enhances safety in busy areas | Management systems need regular updates |

Where the lower limit ($l$), the mode ($m$), and the upper limit ($u$) are three parameters of a TNF.

**Definition 3**: The Graded Mean Integration Representation (GMIR) of $\tilde{a}_i = (l_i, m_i, u_i)$, denoted as $R(\tilde{a}_i)$, is a method used to compute its expected value. Symbolically:

$$R(\tilde{a}_i) = \frac{l_i + 4m_i + u_i}{6}. \tag{2}$$

**3.2.2 Fuzzy BWM.** BWM was developed by Rezaei [15] to help decision-makers identify and prioritize criteria based on numerous pairwise comparisons. Eq (3) reveals how the

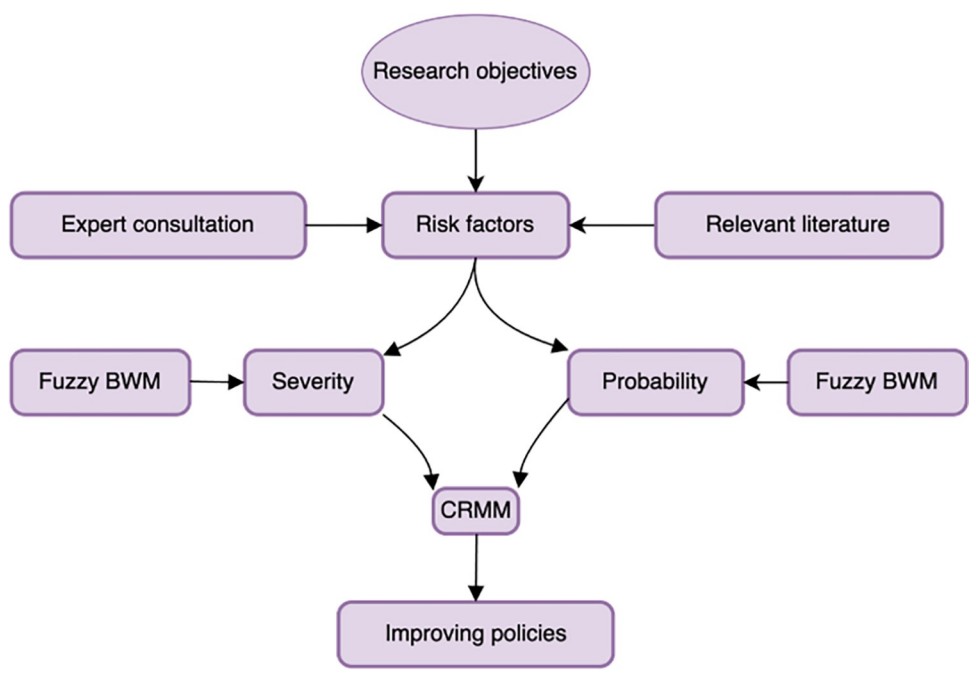

**Fig 1. Research flowchart.**

pairwise comparison matrix A can be formulated for a set of $n$ criteria.

$$A = [a_{ij}] = \begin{bmatrix} a_{11} & a_{12} & \cdots & a_{1n} \\ a_{21} & a_{22} & \cdots & a_{2n} \\ \vdots & \vdots & \ddots & \\ a_{n1} & a_{n2} & \cdots & a_{nn} \end{bmatrix} \tag{3}$$

Where $a_{ij}$ ($i,j = 1,2,\ldots,n$) is defined as a judgment of the $i^{th}$ criterion over the $j^{th}$ one. Besides, the value of $a_{ij}$ can be rated by linguistic terms, as presented in Table 2.

In this paper, the application of BWM can be conducted via the following steps:

**Step 1**: Set up a set of criteria $C = (c_1, c_2, \ldots, c_j, \ldots, c_n)$ for expert judgments.

**Step 2**: Select the best and worst criteria from $n$ criteria, as mentioned in Step 1.

**Step 3**: Conduct fuzzy reference comparisons (FRCs) for the best criterion over all remaining criteria. Consequently, we obtain the fuzzy best vector

(FBV):$\tilde{A}_B = (\tilde{a}_{B1}, \tilde{a}_{B2}, \ldots, \tilde{a}_{Bj}, \ldots, \tilde{a}_{Bn})$. Where $\tilde{a}_{Bj}$ is PFC of the best criterion ($c_B$) over the $j^{th}$ criterion ($c_j$). Note that $\tilde{a}_{BB} = (1, 1, 1)$.

**Step 4**: Perform FRCs for all remaining criteria over the worst criterion. Consequently, we obtain the fuzzy worst vector (FWV):$\tilde{A}_W = (\tilde{a}_{1W}, \tilde{a}_{2W}, \ldots, \tilde{a}_{jW}, \ldots, \tilde{a}_{nW})$. Where $\tilde{a}_{jW}$ is PFC of the $j^{th}$ criterion ($c_j$) over the worst criterion ($c_W$). Note that $\tilde{a}_{WW} = (1, 1, 1)$.

**Step 5**: Determine the optimal fuzzy weights (OFWs) of criteria $\tilde{W}^* = (\tilde{w}_1^*, \tilde{w}_2^*, \ldots, \tilde{w}_j^*, \ldots, \tilde{w}_n^*)$. We expect to find $\tilde{W}^*$ on the condition that for each fuzzy

**Table 2. Consistency index for fuzzy judgments.**

| Linguistic terms | Equally probabilistic (severe) | Weakly probabilistic (severe) | Fairly probabilistic (severe) | Very probabilistic (severe) | Absolutely probabilistic (severe) |
|---|---|---|---|---|---|
| $a_{BW}$ | (1, 1, 1) | (2/3, 1, 3/2) | (3/2, 2, 5/2) | (5/2, 3, 7/2) | (7/2, 4, 9/2) |
| CI | 3 | 3.8 | 5.29 | 6.69 | 8.08 |

pair $\tilde{w}_B/\tilde{w}_j$ and $\tilde{w}_j/\tilde{w}_W$, we have $\tilde{w}_B/\tilde{w}_j = \tilde{a}_{Bj}$ and $\tilde{w}_j/\tilde{w}_W = \tilde{a}_{jW}$. Stated differently, we will obtain an optimal solution where the maximum absolute distance $|\tilde{w}_B/\tilde{w}_j - \tilde{a}_{Bj}|$ and $|\tilde{w}_j/\tilde{w}_W - \tilde{a}_{jW}|$ is minimized for all $j$. To accomplish that, we construct the following constrained optimization [52]:

$$min\ max(j)\left\{\left|\frac{\tilde{w}_B}{\tilde{w}_j} - \tilde{a}_{Bj}\right|, \left|\frac{\tilde{w}_j}{\tilde{w}_W} - \tilde{a}_{jW}\right|\right\}$$

$$S.t:$$

$$\begin{cases} \sum_{j=1}^{n} R(\tilde{w}_j) = 1 \\ l_j^w \le m_j^w \le u_j^w \\ l_j^w \ge 0 \\ j = 1, 2, \dots, n \end{cases} \tag{4}$$

Let $\tilde{\varepsilon} = (l^\varepsilon, m^\varepsilon, u^\varepsilon)$, Model (4) can be transformed to Model (5):

$$min\ \tilde{\varepsilon}$$

$$S.t:$$

$$\begin{cases} \sum_{j=1}^{n} R(\tilde{w}_j) = 1 \\ \left|\frac{\tilde{w}_B}{\tilde{w}_j} - \tilde{a}_{Bj}\right| \le \tilde{\varepsilon} \\ \left|\frac{\tilde{w}_j}{\tilde{w}_W} - \tilde{a}_{jW}\right| \le \tilde{\varepsilon} \\ l_j^w \le m_j^w \le u_j^w, l_j^w \ge 0, j = 1, 2, \dots, n \end{cases} \tag{5}$$

Suppose that $\tilde{\varepsilon}^* = (t^*, t^*, t^*), t^* \le l^\varepsilon$, and let $\tilde{w}_B = (l_B^w, m_B^w, u_B^w)$, $\tilde{w}_j = (l_j^w, m_j^w, u_j^w), \tilde{w}_W = (l_W^w, m_W^w, u_W^w), \tilde{a}_{Bj} = (l_{Bj}, m_{Bj}, u_{Bj})$, and $\tilde{a}_{jW} = (l_{jW}, m_{jW}, u_{jW})$, Model (5) can be transferred to Model (6):

$$min\ \tilde{\varepsilon}^*$$

$$S.t:$$

$$\begin{cases} \sum_{j=1}^{n} R(\tilde{w}_j) = 1 \\ \left|\frac{(l_B^w, m_B^w, u_B^w)}{(l_j^w, m_j^w, u_j^w)} - \left(l_{Bj}, m_{Bj}, u_{Bj}\right)\right| \le (t^*, t^*, t^*) \\ \left|\frac{(l_j^w, m_j^w, u_j^w)}{(l_W^w, m_W^w, u_W^w)} - \left(l_{jW}, m_{jW}, u_{jW}\right)\right| \le (t^*, t^*, t^*) \\ l_j^w \le m_j^w \le u_j^w, l_j^w \ge 0, j = 1, 2, \dots, n \end{cases} \tag{6}$$

Solving Model (6), we obtain OFWs $\tilde{W}^* = (\tilde{w}_1^*, \tilde{w}_2^*, \dots, \tilde{w}_j^*, \dots, \tilde{w}_n^*)$ and $\tilde{\varepsilon}^* = (t^*, t^*, t^*)$.

**Step 6**: Check the consistency of experts' judgment by the consistency ratio (CR):

$$CR = \frac{\tilde{\varepsilon}}{CI} \times 100\% \tag{7}$$

Where CI is the consistency index, whose values are shown in the last row of Table 2. It is argued that $CR < 10\%$ is acceptably consistent [52, 53].

**Step 7**: Combine the individual OFWs. Call $\tilde{W}^{*e} = (\tilde{w}_1^{*e}, \tilde{w}_2^{*e}, \ldots, \tilde{w}_j^{*e}, \ldots, \tilde{w}_n^{*e})$ be the OFWs rated by the $e^{th}$ expert, and $e = (1,2,\ldots,E)$ is a set of experts in the survey. Then, $\tilde{w}_j^*$ can be combined as:

$$w_j^* = \frac{\sum_{e=1}^{E} w_j^{*e}}{E}, j = 1, 2, \ldots, n. \tag{8}$$

## 3.3 Continuous risk management matrix

Different from TRMM, CRMM enables decision-makers to assign continuous values to the probability and severity of RFs, resulting in a more detailed and precise assessment [54]. Additionally, employing continuous scales for these parameters allows CRMM to enhance the consistency of risk factor evaluation via eliminating the arbitrary categorization associated with discrete numbers [55].

In risk management, risk value (RV) can be computed by multiplying the probability that a risk occurs by its severity [9, 41]. In practice, RV is calculated to classify risks, allowing firms to allocate their limited resources to reduce the most impactful risks. By that idea, let $p_i$ and $s_i$ be probability and severity of $RF_i$ ($i = 1,2,\ldots,n$), respectively. Then, RV of $RF_i$ can be determined by Eq (9). From such risk values, we can attain a continuous risk management matrix of risk factors.

$$RV_i = \frac{p_i \times s_i}{\sum_{i=1}^{n} p_i \times s_i} \times 100\% \tag{9}$$

## 4. Empirical application

### 4.1 Hierarchical structure of risk factors for fast-ferry transportation

This paper aims to conduct the navigational risk evaluation of ferry transport by CRMM based on fuzzy BWM. Accordingly, the first is to set up the hierarchical structure of risk factors for ferry transportation. In doing so, the paper relies on the extensive literature review, as done in Section 2, and industrial experts' consultations from the FO-VN case. As a result, the hierarchical structure of RFs of navigational safety for ferry transportation includes five dimensions with 20 RFs, as shown in Table 3.

### 4.2 Data collection

To verify the proposed research model, this paper first selected the three biggest FOs in the South of Vietnam (hereafter the FO-VN case). Next, the current research asked each ferry operator to provide 10~12 officers and senior crew members to interview. More crucially, the risk evaluation in ferry transport is highly professional; thus, selected respondents must have enough knowledge of marine navigation. Then, we designed the expert questionnaire to capture experts' perceptions of RFs' probability and severity. From Table 3, the designed questionnaire includes five dimensions and twenty RFs. Finally, we interviewed respondents face-to-face and by phone (emails) and got 28 valid responses, and the background of which is shown in Table 4.

**Table 3. RFs' hierarchical structure.**

| Layer 1: Dimensions | Layer 2: RFs | Code | Explanation | Sources |
|---|---|---|---|---|
| Human factors (HF) | Crew fatigue | HF1 | Fatigue is a physiological condition characterized by diminished cognitive or physical performance capability. It can be caused by factors such as sleep loss or excessive mental and physical activity. For crew members, fatigue can hinder their ability to operate a ship safely or carry out safety-related responsibilities. | Yuan, Wang [18], Xue, Papadimitriou [8] |
| | Crew unfamiliarity with vessel systems | HF2 | Each vessel has peculiarities which the crew onboard must excel. Unfamiliarity with the ferry's systems can cause navigational misjudgments, delays in response, and ship groundings | Mohammed, Benson [25], Hiremath, Pandey [26] |
| | Insufficient training for crew | HF3 | Lack of crew training can increase the risk of navigation errors and accidents. For instance, insufficient knowledge of safety procedures, and communication systems can compromise the ability of the crew to operate a vessel effectively. | Uğurlu, Yıldırım [20], Hasanspahić, Frančić [21] |
| | Communication breakdown | HF4 | A communication breakdown refers to a failure or interruption in the process of exchanging information among crew onboard, caused by misinterpretation, miscommunication, or even lack of communication. This risk factor can lead to navigational errors, delays in emergency responses, and coordination issues during routine operations and crisis situations. | Nguyen, Ngo [1], Wang, Liu [43] |
| Navigational equipment (NE) | Malfunction of electronic navigation systems | NE1 | Electronic navigation systems, such as GPS (Global Positioning System), radar, and Electronic Chart Display and Information Systems (ECDIS), can experience malfunctions or failures. If not promptly addressed, a failure in these systems can reduce the ferry's ability to navigate accurately, thereby increasing the risk of collisions or grounding. | Hsu, Tai [7], Iperen [64] |
| | Outdated nautical charts | NE2 | Outdated nautical charts often present seabed topography, new navigational hazards, and precise alterations to shipping lanes. Accordingly, relying on such incorrect information can cause some navigational errors, such as miscalculating distances, misjudging water depths, and misunderstanding the layout of the waterway. | Xue, Papadimitriou [8], Mohammed, Benson [25], Hiremath, Pandey [26] |
| | Poor visibility of navigational aids | NE3 | Navigational aids (i.e., buoys, radio beacons, fog signals, and lightships) are used to provide "street" signs on the water for ships. This equipment assists FOs to navigate in the vast ocean where landmarks are not visible, thereby ensuring maritime safety. | Wang, Liu [27], Hiremath, Pandey [26] |
| | Lack of redundancy and back-up systems | NE4 | Redundancy is extra components of a vessel, which is used in case of failure in other elements. Thence, this redundancy is crucial for maintaining the vessel's ability to continue its operations, especially during emergency situations. | Wang and Chin [28], Mia, Uddin [29] |
| Port navigation regulations (PR) | Non-compliance with COLREGs | PR1 | COLREGs are a set of regulations developed by IMO to establish international standards for the safe navigation of ships and prevent collision at sea. Accordingly, non-compliance of these rules can result in an increase in the probability of collisions, and compromise the overall safety of ferry operations. | Başhan, Demirel [31], Ung [32], Arof and Nair [33] |
| | Inadequate implementation of local navigation rules | PR2 | Besides international rules, navigators must adhere to local navigation rules, often in specific waterways of countries and territories. Hence, non-compliance with these rules can generate navigational errors, especially in congested or restricted areas and may cause hazards, such as collisions, grounding, stranding, and even legal actions. | Fan, Wang [34], Hiremath, Pandey [26] |
| | Insufficient training on navigational regulations | PR3 | As mentioned above, following maritime rules and regulations ensure safe navigation and avoid collisions. Thence, insufficient training on these regulations is a leading cause of maritime accidents and injuries. | Baldauf and Hong [35], Hsu, Tai [7] |
| | Poor communication with maritime authorities | PR4 | Effective communication with maritime authorities is essential for safe and efficient navigation operations, and ensuring compliance with regulations. Therefore, poor communication with local authorities can cause severe problems in navigation safety. For instance, the vessel cannot be located near a place where rescue is possible without proper communication. | Başhan, Demirel [31], Ung [32], Arof and Nair [33] |

*(Continued)*

**Table 3.** (Continued)

| Layer 1: Dimensions | Layer 2: RFs | Code | Explanation | Sources |
|---|---|---|---|---|
| Vessel design (VD) | Inadequate stability and seakeeping characteristics | VD1 | Stability and seakeeping characteristics are features of a vessel impacting its ability to remain at sea in all conditions and carry out its intended mission. These characteristics include strength, maneuverability, endurance, and the motions of the vessels. Thus, insufficient stability and seakeeping characteristics can lead to the risk of capsizing, especially in adverse weather conditions (i.e., lightning, thunderstorms, tornadoes), affecting passenger comfort and generating a safety hazard. | Ozturk and Cicek [36], Hsu, Tai [7] |
| | Poor visibility from the bridge | VD2 | The bridge is one of the most important parts of a ship, where its navigation is carried out. So, inadequate visibility from the bridge can compromise the ability of the crew onboard to navigate the boat safely, increasing the probability of maritime accidents. | Yuan, Wang [18], Wang, Liu [43] |
| | Lack of redundancy in propulsion systems | VD3 | Propulsion systems help propel ships through the water. Hence, when primary propulsion systems are damaged, a vessel without redundancy in propulsion systems may face challenges, such as power loss, and operation downtime. | Caris, Limbourg [65], Wood, Collier [40] |
| | Inadequate evacuation and emergency response features | VD4 | Marine transport is one of the most dangerous industries; thus, evacuation and emergency response features are critical parts of vessel design to ensure the safety of vessels, passengers, and crew in the event of emergent situations. Therefore, insufficiency of these features in vessel design can reduce the ability to rescue and evacuate victims in marine accidents. | Akyuz [39], Ung [41] |
| External environment (EE) | Bad weathers | EE1 | Bad weather (i.e., storms, high winds, heavy rain, and rough seas) can negatively impact ship navigation, causing abnormal maneuverability, wrong direction navigation, and cargo damage. | Sys, Van de Voorde [44], Wang, Liu [43] |
| | Strong tidal currents | EE2 | Strong tidal currents and unpredictable water flow patterns are natural phenomena affecting the navigation safe of ferries, especially in coastal and narrow waterways. Additionally, they can make ferries challenging to navigate in the intended course. | Cui [42], Wang, Liu [43] |
| | Floating debris | EE3 | Floating debris refers to objects floating below the surface of water bodies and is often challenging to see in the ocean. Some marine debris (i.e., abandoned and derelict ships) cause vessel damage and hazards to navigation. | Kulkarni, Goerlandt [45], Chang, Xu [46] |
| | Marine traffic congestion | EE4 | Marine traffic congestion refers to the scenario in which multiple ships share the same water space. It is argued that a high density of vessels in congested waterways can increase the probability of collisions. | Xu, Ma [48], Wang, Liu [43] |

Table 4 presents the demographic background of respondents. Marine engineers, catering staff, and deck officers constitute the largest job title category (78.57%). Meanwhile, most respondents (75%) have 5 to 20 years of seniority. Besides, 75% of respondents hold undergraduate degrees, and the age distribution is relatively balanced, with the largest group falling within the 36–46 years range. Most crucially, 100% of respondents have a safety license in marine navigation issued by the Vietnam Maritime Administration.

## 4.3 Probability and severity weight of RFs

As discussed above, the probability and severity weights of RFs are calculated by fuzzy BWM, as seen from Steps (1) ~ (7). Initially, the current paper determines the global weights of five main dimensions in Layer 1. Then, the local weights of RFs in Layer 2 corresponding to each dimension are computed. Finally, the global weights of RFs are figured out by multiplying the global weights of such five dimensions with the local weights of RFs. The result for the FO-VN case is shown in Table 5. It is evident that three RFs with higher probability weight include

**Table 4. Respondents' background.**

| Features | | Frequency | % |
|---|---|---|---|
| Job titles | Master | 3 | 10.71 |
| | Operations Manager | 3 | 10.71 |
| | Deck Officer | 6 | 21.43 |
| | Catering Staff | 7 | 25.00 |
| | Marine Engineer | 9 | 32.14 |
| Seniority (year) | 5~10 | 11 | 39.29 |
| | 11~20 | 10 | 35.71 |
| | Over 20 | 7 | 25.00 |
| Education | Undergraduate | 21 | 75.00 |
| | Postgraduate | 7 | 25.00 |
| Age (year) | 25~35 | 6 | 21.43 |
| | 36~46 | 13 | 46.43 |
| | Over 46 | 9 | 32.14 |
| Safety license | Yes | 28 | 100.00 |
| | No | 0 | 0.00 |

HF4 (8.99%), PR4 (8.43%), and EE4 (8.27%). Meanwhile, three RFs with higher severity weight comprise VD4 (15.59%), VD1 (11.98%), and PR3 (7.21%).

## 4.4 Continuous risk matrix

Based on the probability and severity weights of RFs, as computed in Section 4.3, applying Eq (9), the risk value of RFs is found and exhibited in the second-to-last column of Table 6. After

**Table 5. RFs' probability and severity weight.**

| Layer 1: Dimensions | Global weight of dimensions | | Layer 2: RFs | Local weight of RFs | | Global weight of RFs | |
|---|---|---|---|---|---|---|---|
| | Probability | Severity | | Probability | Severity | Probability | Severity |
| Human factors (HF) | 23.84 | 6.16 | HF1 | 13.02 | 17.58 | 3.10 | 1.08 |
| | | | HF2 | 26.57 | 15.69 | 6.33 | 0.97 |
| | | | HF3 | 22.71 | 38.44 | 5.41 | 2.37 |
| | | | HF4 | 37.71 | 28.29 | 8.99 | 1.74 |
| Navigational Equipment (NE) | 15.59 | 13.51 | NE1 | 36.88 | 25.33 | 5.75 | 3.42 |
| | | | NE2 | 15.45 | 16.95 | 2.41 | 2.29 |
| | | | NE3 | 24.85 | 24.86 | 3.88 | 3.36 |
| | | | NE4 | 22.82 | 32.87 | 3.56 | 4.44 |
| Port Navigation Regulations (PR) | 24.84 | 18.06 | PR1 | 21.01 | 16.55 | 5.22 | 2.99 |
| | | | PR2 | 18.64 | 17.74 | 4.63 | 3.20 |
| | | | PR3 | 26.41 | 39.90 | 6.56 | 7.21 |
| | | | PR4 | 33.94 | 25.81 | 8.43 | 4.66 |
| Vessel Design (VD) | 14.83 | 42.40 | VD1 | 18.07 | 28.26 | 2.68 | 11.98 |
| | | | VD2 | 25.33 | 19.55 | 3.76 | 8.29 |
| | | | VD3 | 31.26 | 15.42 | 4.63 | 6.54 |
| | | | VD4 | 25.35 | 36.77 | 3.76 | 15.59 |
| External Environment (EE) | 20.90 | 19.87 | EE1 | 14.87 | 18.16 | 3.11 | 3.61 |
| | | | EE2 | 21.52 | 35.25 | 4.50 | 7.00 |
| | | | EE3 | 24.05 | 14.07 | 5.03 | 2.79 |
| | | | EE4 | 39.56 | 32.53 | 8.27 | 6.46 |

**Table 6. RFs' risk values.**

| RFs | Probability | Severity | RV (%) | Category |
|---|---|---|---|---|
| VD4 | 3.76 | 15.59 | 12.43 | Extreme (E) |
| EE4 | 8.27 | 6.46 | 11.34 | |
| PR3 | 6.56 | 7.21 | 10.03 | |
| PR4 | 8.43 | 4.66 | 8.34 | High (H) |
| VD1 | 2.68 | 11.98 | 6.81 | |
| EE2 | 4.50 | 7.00 | 6.68 | |
| VD2 | 3.76 | 8.29 | 6.60 | |
| VD3 | 4.63 | 6.54 | 6.43 | |
| NE1 | 5.75 | 3.42 | 4.17 | Medium (M) |
| NE4 | 3.56 | 4.44 | 3.35 | |
| HF4 | 8.99 | 1.74 | 3.33 | |
| PR1 | 5.22 | 2.99 | 3.31 | |
| PR2 | 4.63 | 3.20 | 3.15 | |
| EE3 | 5.03 | 2.79 | 2.98 | |
| NE3 | 3.88 | 3.36 | 2.76 | |
| HF3 | 5.41 | 2.37 | 2.72 | |
| EE1 | 3.11 | 3.61 | 2.38 | Low (L) |
| HF2 | 6.33 | 0.97 | 1.30 | |
| NE2 | 2.41 | 2.29 | 1.17 | |
| HF1 | 3.10 | 1.08 | 0.71 | |

that, RFs are divided into four categories. Moreover, this study utilizes the "ggRepel" package in Rstudio to visually represent CRMM. As shown in Fig 2, CRMM presents the probability and severity weights on the horizontal and vertical axis, respectively. Evidently, CRMM classifies RFs into four risk areas. In particular, the extreme-risk area includes three RFs, such as VD4 (12.43%), EE4 (11.34%), and PR3 (10.03%). The high-risk area consists of five RFs, for example, PR4 (8.34%), VD1 (6.81%), EE2 (6.68%), VD2 (6.60%), and VD3 (6.43%). In addition, eight RFs are located in the medium-risk area, while four RFs are situated in the low-risk zone. It has been posited that for the risk management process, FOs should prioritize the extreme-risk RFs in the context of limited resources [18, 45, 56].

## 4.5 Discussion

Three RFs in the FO-VN case, namely, inadequate evacuation and emergency response features (VD4), marine traffic congestion (EE4), and insufficient training on navigational regulations (PR3), are identified as having an extreme-risk level through CRMM. From risk management perspectives, FOs are advised to prioritize their attention to such RFs, particularly in the case of limited resources. From these empirical findings and a comprehensive literature review, the authors conducted post-interviews with professional experts from the survey, and some managerial recommendations to improve the safety of navigation for ferry transportation are suggested as follows:

It is argued that inadequate evacuation and emergency response features are risks associated with the ferry design [39, 41]. These features are designed to ensure the safety of passengers and crew members during emergencies and evacuations on ferry vessels [46]. Thus, inadequate emergency response features (i.e., firefighting systems and communication tools) may escalate the severity of incidents and compromise the safety of passenger and crew on board [57, 58]. According to the expert interviewed, all ferry crews must take part in regular

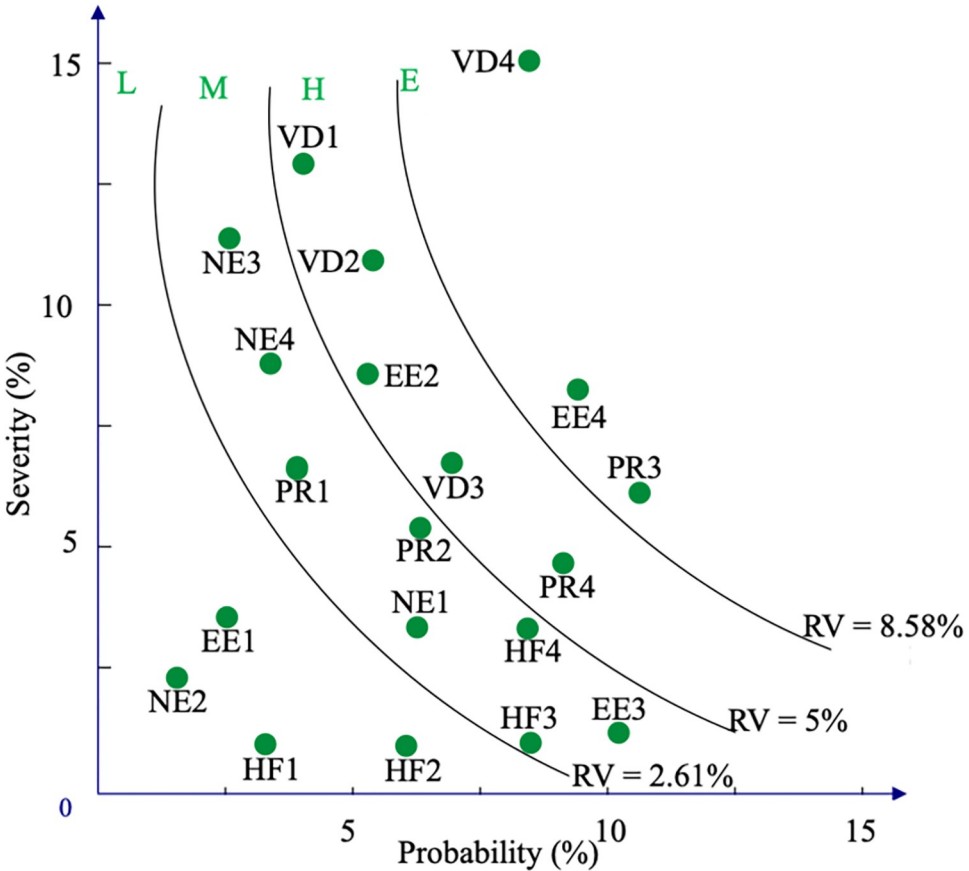

**Fig 2. The visualization of CRMM for the FO-VN case.**

emergency drills to improve their skills during the crisis. Furthermore, these drills should represent practical emergency scenarios in ferry navigation, such as fire, grounding, collision, etc., to ensure that they are familiar with evacuation processes. This suggestion is in line with Kim and Moon [56], and Kulkarni, Goerlandt [45]. In addition, ferries are also suggested to be equipped with modern life-saving equipment, such as home defibrillators, lifeboats, and inflatable buoyant apparatus for safety navigation.

Marine traffic congestion is arguably an extreme-risk factor in the FO-VN case that should be reduced to improve the navigation safety for ferry transport. According to Yuan, Wang [18], congested waterways hinder free movement and cause trouble in the safe maneuvering of ships, thus increasing the likelihood of collisions between ferries and other vessels. The interviewed experts reached a high consensus that to reduce congestion in marine traffic, FOs should use advanced technology such as Automatic Identification System (AIS), radar, and satellite tracking to monitor vessel movements in real-time. A Long Short-Term Memory (LSTM) neural network model is also studied at Shanghai, Singapore and Ningbo ports for congestion and sequence prediction [59]. Another policy is to boost collaboration and communication between FOs and port operators (POs) to alleviate marine traffic congestion. Nonetheless, this policy can be possible if all parties (i.e., FOs and POs) have access to the exact source of the newest information. This suggestion is relatively consistent with Hsu, Tai [7], and Nguyen, Ngo [1].

Navigational regulations for ships are rules and guidelines established to ensure the safe and efficient navigation of vessels at sea [8, 60]. Accordingly, inadequate training for FOs and

crew members in understanding and adhering to these regulations can create many risks for ferry navigation safety, such as unsafe maneuvers [35], misinterpretation of charts [20], and the increase in the probability of collision [32]. Based on empirical findings, interviewed experts suggest the following two policies to boost this issue. The first policy is to design courses including all critical aspects of maritime navigational regulations, such as COLREGs, the International Convention for the Prevention of Pollution from Ships (MARPOL), the Pilotage Act, etc. After finishing these courses, ferry personnel can be certified and continuously updated on relevant navigation regulations. The second one is to assess crew members on the completion of their training to demonstrate that they can apply navigation regulations in real-world cases. This recommendation is in agreement with Xue, Papadimitriou [8] and Fan, Wang [34].

It is imperative to discuss the main advantages of the proposed methods. Unlike TRMM, CRMM allows decision-makers to assign continuous values to the probability and severity of RFs; thus, RFs are assessed more detailed and precise. Besides, by using continuous scales of the probability and severity of RFs, CRMM improves consistency in RFs' evaluation thanks to avoiding the arbitrary categorization of risks into discrete numbers. Moreover, fuzzy BWM requires less PCs, thus minimizing the inconsistency that is often encountered in other methods, for instance AHP, ANP, and SAW. Additionally, the incorporation of fuzzy logic in BWM allows to handle uncertainty and subjectivity of DMs' ratings. Thence, it would be said that fuzzy BWM is particularly useful in evaluation RFs in ferry transport.

To conclude, using CRMM, FOs are advised to prioritize their attention to the extreme-risk factors, particularly in resource-limited situations. Based on that, the experts in the empirical case suggest many policies for FOs to mitigate the identified the extreme-risk factors. These strategies encompass enhancing emergency response features, leveraging technology to manage marine traffic congestion, and prioritizing comprehensive training courses for crew members to ensure that they adhere to navigational regulations.

## 5. Conclusion

This paper aims to conduct a navigational risk evaluation of ferry transport by CRMM based on fuzzy BWM. Some theoretical and practical contributions can be addressed as follows:

First, from the literature review and ferry navigation's feature, the current paper identifies five dimensions with twenty RFs for ferry navigation. By means of CRMM, RFs are divided into four zones corresponding to their risk values, including extreme-risk, high-risk, medium-risk, and low-risk areas. Thanks to that, DMs can make policies to allocate resources to improve the safety of ferry navigation. It is argued that under the circumstance of limited resources, FOs should prioritize to allocate the resources to RFs in the extreme-risk area. By contrast, resources being used for RFs in the low-risk area should be deployed elsewhere, preferably RFs in the extreme- and high-risk areas.

Second, the application of fuzzy BWM in calculating the probability and severity of RFs can provide a methodological reference to the MCDA research. Compared with the pairwise comparison-based tools (i.e., AHP, ANP, and SAW), BWM does not require a full pairwise comparison matrix [15]. Thus, it needs less data and produces more consistent results [52, 53]. Further, it is argued that judging the probability and severity of RFs is inherently uncertain and subjective. Therefore, integrating fuzzy theory into BWM allows a more realistic representation of uncertainty, vagueness, and imprecision. The proposed method could be extended to various real-life decision-making problems, such as bio-medical waste management [61], plastic waste management [62], renewable energy sources [63].

Third, as an empirical study, the paper surveys three major FOs in Vietnam to verify the proposed research model. Results identify three extreme-risk RFs, comprising inadequate evacuation and emergency response features (VD4), marine traffic congestion (EE4), and insufficient training on navigational regulations (PR3). In practice, these RFs should be given a high priority in resource allocation. Some suggested policies to improve these RFs include enhancing emergency response features, leveraging technology to manage marine traffic congestion, and prioritizing comprehensive training courses to ensure crew have advanced knowledge of maritime regulations.

The current research exists some research limitations, as follows. Initially, the traditional assumption of criteria independence used in BWM makes it unrealistic in many real-world cases. For instance, *crew familiarity with vessel systems* (HF2) and *lack of crew training and competence* (HF3) correlate with each other. Nonetheless, such a correlation is not considered in the study. Therefore, how to revise the weight vector determined by fuzzy BWM is an area of future research. Second, the operation of fuzzy BWM is based on optimization techniques. Accordingly, complex and high-dimensional problems, viz., many criteria and alternatives involved, may pose challenges to the adoption of fuzzy BWM. It is highly recommended that specialized software be developed to employ fuzzy BWM more practically and efficiently.

## Supporting information

**S1 File. Surveyed data.**
(ZIP)

## Acknowledgments

The authors would like to thank colleagues for very thoughtful reviews and critical comments, which have led to significant improvements to the early versions of the manuscript.

## Author Contributions

**Conceptualization:** Long Van Hoang.

**Data curation:** Long Van Hoang.

**Formal analysis:** Long Van Hoang.

**Project administration:** Long Van Hoang.

**Validation:** Linh Thi Pham.

**Visualization:** Linh Thi Pham.

**Writing – original draft:** Linh Thi Pham.

**Writing – review & editing:** Linh Thi Pham.

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
