## [Decision Letter · Decision Letter 0]

30 Apr 2024

PONE-D-24-02990A navigational risk evaluation of ferry transport: Continuous risk management matrix based on fuzzy Best-Worst MethodPLOS ONE

Dear Dr. Hoang,

Thank you for submitting your manuscript to PLOS ONE. After careful consideration, we feel that it has merit but does not fully meet PLOS ONE’s publication criteria as it currently stands. Therefore, we invite you to submit a revised version of the manuscript that addresses the points raised during the review process.

We look forward to receiving your revised manuscript.

Kind regards,

Orhan Ozgur AYBAR, PhD

Academic Editor

PLOS ONE

Journal Requirements:

Additional Editor Comments:

Dear Authors,

Following comments should be addressed to be considered for publication in Plos ONE.

Referee 1

This paper evaluates the navigational risk of ferry transport by a continuous risk management matrix based on the fuzzy Best-Worst Method. I think the authors have written an interesting paper dealing with an important topic. The overall representation of this paper is technically sound. I have, however, a few comments and suggestions for them:

1. The abstract should be modified. It is therefore recommended that the abstract be carefully rewritten to effectively demonstrate the necessity, novelty, and contribution of the research, as well as highlight its major findings. Write the whole abstract in a narrative form rather than itemize.

2. I suggest summarizing in a table with challenges and proposed solutions by the different authors, their strengths and weaknesses. This will immediately make clear the new proposal's need and the evolution that it represents.

3. The motivation for this is not highlighted in the manuscript. The manuscript's value is hidden from the reader. It needs to be emphasized in a paragraph clearly.

4. The main contribution of the manuscript should be listed in bullets under the Introduction section.

5. Why triangular fuzzy best worst method is utilized? Give reasons.

6. It is imperative to compare the proposed method with the literature. The main advantage of the proposed method should be discussed comparatively with other existing solutions.

7. It is noteworthy that your approach your method could be extended to various real-life decision-making problems, such as the ‘bio-medical waste management, plastic waste management, renewable energy sources’, and more. https://doi.org/10.1016/j.eswa.2023.120082;
https://doi.org/10.1016/j.asoc.2023.110516;
https://doi.org/10.1016/j.asoc.2024.111495;Include this in the future scope of the conclusion section with proper citation of relevant references.

Referee 2

1. The paper is of importance in terms of merging continuous risk matrix and Fuzzy BWM. But I wonder how the authors generate the Figure 2. I think more details needed to explain how they structured the x-and y-axis values. What is the continuous function here?

2. Going on from the end of comment 1, how they determine the four categories? Why four?

3. Why the authors prefer fuzzy version of BWM? Is there a special reason or not? Please explain in detail.

4. I think a flow chart is required to easily understand the integration between CRMM and fuzzy BWM. More theoretical information is needed for CRMM.

5. Authors provided a consistency computation in their method part but there is no output for this in the application. Please provide your results regarding consistency.

Reviewers' comments:

Reviewer's Responses to Questions

**Comments to the Author**

1. Is the manuscript technically sound, and do the data support the conclusions?

Reviewer #1: Partly

Reviewer #2: Yes

2. Has the statistical analysis been performed appropriately and rigorously? 

Reviewer #1: N/A

Reviewer #2: Yes

3. Have the authors made all data underlying the findings in their manuscript fully available?

Reviewer #1: Yes

Reviewer #2: Yes

4. Is the manuscript presented in an intelligible fashion and written in standard English?

Reviewer #1: Yes

Reviewer #2: Yes

5. Review Comments to the Author

Reviewer #1: This paper evaluates the navigational risk of ferry transport by a continuous risk management matrix based on the fuzzy Best-Worst Method. I think the authors have written an interesting paper dealing with an important topic. The overall representation of this paper is technically sound. I have, however, a few comments and suggestions for them:

1. The abstract should be modified. It is therefore recommended that the abstract be carefully rewritten to effectively demonstrate the necessity, novelty, and contribution of the research, as well as highlight its major findings. Write the whole abstract in a narrative form rather than itemize.

2. I suggest summarizing in a table with challenges and proposed solutions by the different authors, their strengths and weaknesses. This will immediately make clear the new proposal's need and the evolution that it represents.

3. The motivation for this is not highlighted in the manuscript. The manuscript's value is hidden from the reader. It needs to be emphasized in a paragraph clearly.

4. The main contribution of the manuscript should be listed in bullets under the Introduction section.

5. Why triangular fuzzy best worst method is utilized? Give reasons.

6. It is imperative to compare the proposed method with the literature. The main advantage of the proposed method should be discussed comparatively with other existing solutions.

7. It is noteworthy that your approach your method could be extended to various real-life decision-making problems, such as the ‘bio-medical waste management, plastic waste management, renewable energy sources’, and more. https://doi.org/10.1016/j.eswa.2023.120082;
https://doi.org/10.1016/j.asoc.2023.110516;
https://doi.org/10.1016/j.asoc.2024.111495;Include this in the future scope of the conclusion section with proper citation of relevant references.

Reviewer #2: 1. The paper is of importance in terms of merging continuous risk matrix and Fuzzy BWM. But I wonder how the authors generate the Figure 2. I think more details needed to explain how they structured the x-and y-axis values. What is the continuous function here?

2. Going on from the end of comment 1, how they determine the four categories? Why four?

3. Why the authors prefer fuzzy version of BWM? Is there a special reason or not? Please explain in detail.

4. I think a flow chart is required to easily understand the integration between CRMM and fuzzy BWM. More theoretical information is needed for CRMM.

5. Authors provided a consistency computation in their method part but there is no output for this in the application. Please provide your results regarding consistency.

6. PLOS authors have the option to publish the peer review history of their article (what does this mean?). If published, this will include your full peer review and any attached files.

Reviewer #1: **Yes: **Mijanur Rahaman Seikh

Reviewer #2: No

---

## [Decision Letter · Decision Letter 1]

16 Aug 2024

A navigational risk evaluation of ferry transport: Continuous risk management matrix based on fuzzy Best-Worst Method

PONE-D-24-02990R1

Dear Dr. Hoang,

We’re pleased to inform you that your manuscript has been judged scientifically suitable for publication and will be formally accepted for publication once it meets all outstanding technical requirements.

Kind regards,

Muhammet Gul, Ph.D.

Academic Editor

PLOS ONE

Additional Editor Comments (optional):

Reviewers' comments:

Reviewer's Responses to Questions

**Comments to the Author**

1. If the authors have adequately addressed your comments raised in a previous round of review and you feel that this manuscript is now acceptable for publication, you may indicate that here to bypass the “Comments to the Author” section, enter your conflict of interest statement in the “Confidential to Editor” section, and submit your "Accept" recommendation.

Reviewer #1: All comments have been addressed

Reviewer #2: All comments have been addressed

2. Is the manuscript technically sound, and do the data support the conclusions?

Reviewer #1: Yes

Reviewer #2: Yes

3. Has the statistical analysis been performed appropriately and rigorously? 

Reviewer #1: (No Response)

Reviewer #2: N/A

4. Have the authors made all data underlying the findings in their manuscript fully available?

Reviewer #1: (No Response)

Reviewer #2: Yes

5. Is the manuscript presented in an intelligible fashion and written in standard English?

Reviewer #1: (No Response)

Reviewer #2: Yes

6. Review Comments to the Author

Reviewer #1: I have checked the revised manuscript with track changes and the reply to the reviewers. The authors have necessary changes according to the reviewers comment and improve the manuscript to a great extent. However, some portions of the original manuscript file are unreadble maybe due to technical issues during compilation of files. Kindly, check this issue before publication.

Reviewer #2: Authors made a successful revision considering my and other reviewers' comments. Thus I support and recommend publication in PLOSONE.

7. PLOS authors have the option to publish the peer review history of their article (what does this mean?). If published, this will include your full peer review and any attached files.

Reviewer #1: **Yes: **Mijanur Rahaman Seikh

Reviewer #2: No

---

## [Editor Report · Acceptance letter]

21 Aug 2024

PONE-D-24-02990R1 

PLOS ONE

Dear Dr. Hoang, 

I'm pleased to inform you that your manuscript has been deemed suitable for publication in PLOS ONE. Congratulations! Your manuscript is now being handed over to our production team.

Kind regards, 

on behalf of

Dr. Muhammet Gul 

Academic Editor

PLOS ONE